# Infectious diseases burden and antibiotic prescribing patterns among primary care patients in Harare, Zimbabwe – a cross-sectional analysis

**Ioana D. Olaru**[1,2,3]*, **Rudo M. S. Chingono**[1], **Fadzaishe Mhino**[1], **Celia Gregson**[1,4], **Christian Bottomley**[5], **Tsitsi Bandason**[1], **Chipo E. Mpandaguta**[1], **Karlos Madziva**[1], **Rashida A. Ferrand**[1,2], **Michael Vere**[6], **Prosper Chonzi**[6], **Shungu Munyati**[7], **Justin Dixon**[1,8], **Thomas C. Darton**[9], **Katharina Kranzer**[1,2,10]

1 The Health Research Unit Zimbabwe, Biomedical Research and Training Institute, Harare, Zimbabwe, 2 Clinical Research Department, London School of Hygiene and Tropical Medicine, London, United Kingdom, 3 Institute of Medical Microbiology, University Hospital Münster, Münster, Germany, 4 Global Health and Ageing Research Unit, Bristol Medical School, University of Bristol, Bristol, United Kingdom, 5 Department of Infectious Disease Epidemiology, London School of Hygiene and Tropical Medicine, London, United Kingdom, 6 Department of Health, Harare City Council, Harare, Zimbabwe, 7 Biomedical Research and Training Institute, Harare, Zimbabwe, 8 Department of Global Health and Development, London School of Hygiene and Tropical Medicine, London, United Kingdom, 9 Clinical Infection Research Group, School of Medicine and Population Health, University of Sheffield, Sheffield, United Kingdom, 10 Division of Infectious and Tropical Medicine, Medical Centre of the University of Munich, Munich, Germany

* ioana-diana.olaru@lshtm.ac.uk

## Abstract

Low- and middle-income countries (LMIC) continue to experience a high burden of infectious diseases and disparities in access to and use of antimicrobials, yet data on antibiotic prescribing in outpatient settings, where the majority of global prescriptions occur, remain scarce. The objective of this study is to provide data on diagnoses and antibiotic prescriptions among primary care patients in Harare, Zimbabwe. We conducted a retrospective study of medical records from eight primary care clinics in Harare, Zimbabwe. Clinics were selected based on the population they served and the availability of records. Patient consultations conducted between January 2016 and December 2022 were included. Antibiotic prescriptions were categorised into groups according to the AWaRe (Access, Watch and Reserve) classification. During the study period, 199,880 patient consultations were recorded. The median patient age was 9 years and 52.5% (105,035/199,880) were female. The most common causes of presentation were due to infectious diseases including, in order of frequency, gastroenteritis (15.2%; 30,352/199,880), acute respiratory infections (10.9%; 21,381/199,880) and pneumonia (10.5%; 20,889/199,880). Overall, antibiotics were prescribed in 70.5% (117,674/166,858) of patients who were not referred to hospital. Antibiotics commonly prescribed were amoxicillin (39.4%; 65,825/166,858), ciprofloxacin (10.3%; 17,162/166,858), metronidazole (9.4%; 15,681/166,858). Among those who were prescribed antibiotics and not referred, 70.6% (83,034/117,674) were prescribed 'Access' and 29.3% (34,472/117,674) 'Watch' group antibiotics. Patients with respiratory infections,

**Data availability statement:** Data supporting the findings of this study are available within the article and its supplementary materials. Once data analyses related to this project are complete, the data will be deposited into an online repository.

**Funding:** The study was funded by the Wellcome Trust (Grant number 219736/Z/19/Z). IDO received funding though the Wellcome Trust Clinical PhD Programme awarded to the London School of Hygiene & Tropical Medicine (grant number 203905/Z/16/Z). The funders had no role in study design, data collection and analysis, decision to publish, or preparation of the manuscript.

**Competing interests:** The authors have declared that no competing interests exist.

including those with upper respiratory infections, and gastroenteritis were frequently prescribed antibiotics. This study shows that infectious diseases remain a common reason for primary care presentation and antibiotics were frequently prescribed. These findings highlight the need for increasing access to diagnostics in primary care, and for antibiotic stewardship and other context-adapted interventions aimed at optimising patient management and reducing unnecessary antibiotic prescriptions.

## Introduction

Low- and middle-income countries (LMIC) continue to experience a disproportionately high burden of infectious disease resulting in considerable morbidity and mortality [1]. Globally, antimicrobial use has increased by almost 50% over the last two decades. However, considerable disparities in access and use persist [2]. In the face of increasing antimicrobial resistance (AMR), disparities in access to effective antimicrobials, are likely to have an even greater effect in low-resource settings due to costs and availability.

Over 90% of antibiotics globally are prescribed in outpatient settings [3,4] and most primary care consultations in LMIC result in antibiotic prescriptions [5]. Stewardship programmes can lead to reductions in antibiotic prescribing however their effect in primary care has been modest and variable across settings and patient populations [6,7]. In contrast, lacking access to antibiotics in LMIC can result in excess mortality [8] highlighting the need for nuanced stewardship interventions while ensuring unrestricted access for those in need of antibiotic treatment. This is further complicated by shortages in health care workers resulting in overburdened staff and long waiting times for patients. Limited availability of diagnostics and information on infectious diseases epidemiology results in diagnostic uncertainties and further impacts on prescribing decisions.

Antimicrobial use in the WHO African region remains relatively understudied. Only 15 studies were identified by a systematic review of studies on antimicrobial prescriptions in primary care published between 2010 and 2020, most with relatively small sample sizes [5]. However, antibiotic prescriptions were generally high and wide-ranging from 20 to 88% [5]. This results in limited evidence for informing local policies and interventions aimed at reducing the burden of infections and optimising antibiotic use. Infectious diseases are common in Zimbabwe but data on clinic attendances and antibiotic use are lacking. Zimbabwe also has a high prevalence of HIV of 12.9% among adults [9] while 21.6% and 2.9% children under the age of 5 have either stunting or wasting [10], all of which increase the risk for and severity of infections.

The objective of this study is to describe diagnoses and antibiotic prescriptions among adults and children presenting to primary care in Harare, Zimbabwe.

## Methods

### Study design and setting

We performed a retrospective study of medical records from eight primary care clinics in Harare, Zimbabwe (S1 Fig A in S1 File shows a map of the included clinics). Clinics were selected based on their catchment population and on the availability of medical records. In Harare, primary care clinics are public outpatient health facilities providing care to patients of all ages. The clinics are located in eight high-density areas and serve a low-income population. The clinics act as a first point of contact with a health care provider for these communities. According to the latest national census from 2022, the clinics have a combined catchment

population of >705,000 representing almost half of the population of Harare [11]. Primary care is provided by trained nurses who conduct consultations and prescribe treatment for a wide range of conditions including communicable and non-communicable diseases. Patients with severe conditions are recommended to refer to hospital for further evaluation and care. Consultation fees are charged for adults, but they are free of charge for children under the age of five years, and for older people age ≥65 years. Consultation fees impact access to health care with individuals required to pay being less likely to present for minor conditions. Consultations are recorded using paper registers kept at the clinics.

## Data collection

Data on all recorded patient consultations from the outpatient department conducted between 1st January 2016 and 31st December 2022 were collected from clinic paper registries. Patients presenting to the tuberculosis and HIV clinics as well as those from the antenatal-, maternity and well-child clinics were excluded. Data were collected retrospectively between 6th January 2020 and 16th April 2022 and included age, sex, diagnoses and/or symptoms, treatment prescribed, and whether a hospital referral was made. Data were collected using electronic forms in Open Data Kit (ODK), including pre-coded options for common diagnoses and treatment prescribed, with the option to also enter data as free text should the categories not apply. Data were cleaned, checked for consistency, and recoded into diagnostic categories by a medical doctor. Diagnoses, including those entered as free text, were reclassified into broader categories. Because the range of diagnoses recorded was relatively limited, typographical errors could be easily identified and corrected. Respiratory tract infections were defined as one of the following: upper respiratory tract infections (URTI), lower respiratory tract infections (LRTI), respiratory tract infections (RTI); acute respiratory infections (ARI), bronchitis, pneumonia, sinusitis, pharyngitis, tonsilitis, otitis, and COVID-19. Diagnoses which were less common (<2%) were categorised as "other". Fever was defined as a temperature of 37.5°C and above, as documented in the records.

Antibiotic prescriptions were categorised into groups according to the AWaRe (Access, Watch and Reserve) classification [12]. Antibiotics belong to the 'Access' group if they are narrow-spectrum and have a good safety profile. 'Watch' antibiotics have a broader spectrum and a higher resistance potential, while 'Reserve' antibiotics are last-choice used to treat multidrug-resistant infections [13]. Antibiotics analysed were all systemic medicines (oral or injectable) with antibacterial activity, except those used for tuberculosis treatment. Antiviral, antifungal, antiparasitic, and topical drugs were not considered. Clinic presentations according to age were compared to the population age structure from the latest census data from 2022 [11]. While the clinic paper records contained the full patient information, only anonymised data were entered in the electronic database and individuals were not identifiable to researchers accessing the database or conducting the analysis.

## Statistical analysis

The population included was as follows: all patients presenting to the outpatient department for the analysis of diagnoses, and all patients minus those who were directly referred to hospital or for whom the referral status was unknown for antibiotic prescriptions (irrespective of antibiotics receipt prior to referral). This approach was used because antibiotics may be prescribed upon hospital admission among those referred. Categorical variables were presented as counts and percentages and continuous variables as medians and interquartile ranges (IQR). Unadjusted odds ratios and 95% confidence intervals were computed. Diagnoses were presented according to their frequency. Data analysis and visualisations were conducted in R version 4.4.1.

### Ethical considerations

The study was approved by the London School of Hygiene & Tropical Medicine (Ref. 16424) and Medical Research Council of Zimbabwe (MRCZ/A/2406) ethics committees. This was a retrospective study of routine medical records and the requirement for individual consent was therefore waived.

## Results

### Patient characteristics and diagnoses

During the 7-year study period, 199,880 patient consultations were recorded across the eight clinics (S1 Table A in S1 File). Table 1 presents main patient characteristics and diagnoses. The median patient age was 9 (IQR 1.8-32.0) years and 52.5% were female. When comparing the age distribution of those presenting to primary care clinics with the general population [11], children under the age of 5 years were overrepresented (comprising 41% of the clinic presentations) while 10–20-year-olds were underrepresented (S1 Fig B in S1 File shows the age distribution for patients presenting to the clinic compared to census data).

The most common diagnoses in the primary care clinics were gastroenteritis (15.2%; 30,352/199,880), followed by acute respiratory infections (10.9%; 21,381/199,880), pneumonia (10.5%; 20,889/199,880), tonsilitis (7.0%; 13,940/199,880), sexually transmitted infections (5.3%; 10,587/199,880), skin- and soft tissue infections (3.9%; 7,861/199,880), and upper respiratory tract infections (3.9%; 7,777/199,880). Respiratory tract infections and gastrointestinal infections were very common in young children (Fig 1) comprising 48.4% (38,342/79,149) and 19.5% (15,456/79,149) of attendances in children under five. However, presentations for gastroenteritis remained relatively common throughout the life course (Fig 1). Sexually transmitted infections were most prevalent among young adults (15.9%; 7,969/50,106 among 20–39-year-olds) compared to other age groups. Frequency of non-communicable conditions (hypertension and musculoskeletal pain) increased with age. Diabetes mellitus and congestive heart failure were rarely recorded in 0.2% (320/199,880) and 0.1% (133/199,880) of patients, respectively. The distribution of diagnoses was largely similar between men and women with trauma being more common in men and hypertension in women (Fig 1C, Fig 1D, S1 Fig C in S1 File).

### Referrals to hospital

One in seven patients (14.9%; 29,141/195,999) were referred to hospital (Table 1). Referrals did not differ substantially by sex. Among those who were referred to hospital, common conditions were traumatic events (18.9%; 5,509/29,141), pneumonia (9.2%; 2,673/29,141), skin and soft tissue infections (6.6%; 1,918/29,141), and hypertension (6.0%; 1,753/29,141, S1 Fig D in S1 File presents a comparison of main diagnoses resulting in referrals by sex). Patients diagnosed with gastrointestinal infections were referred in 4.3% (1,256/29,141) of cases; upper respiratory tract infections accounted for 0.4% (114/29,141) of hospital referrals. The proportion of recommended referrals was similar across age-groups for gastro-intestinal infection (1.3–5.4%) and pneumonia (6.7–14.4%) apart from children <5 years who were more frequently referred for both pneumonia (14.4%; 2,055/14,238) and gastrointestinal infections (4.9%; 751/15,307; S1 Fig E in S1 File).

### Antibiotic prescriptions

Among those who were not referred to hospital, antibiotics were prescribed in 70.5% (117,674/166,858) of patients. Antibiotics most commonly prescribed were amoxicillin

**Table 1. Characteristics of patients presenting to primary care in eight clinics from Harare, Zimbabwe.**

| Characteristic | Total N=199 880 | Discharged home | | Referred to hospital N=29 141 | Unknown referral status N=3881 |
|---|---|---|---|---|---|
| | | Antibiotics prescribed N=117 674 | No antibiotics prescribed N=49 184 | | |
| Age, median (IQR) years§ | 9 (2–32) | 6 (2–28) | 18 (2–38) | 20 (2–36) | 20 (2–42) |
| **Age group (years), n (%)** | | | | | |
| 0–4 | 79 149 (41.0) | 50 340 (44.2) | 18 272 (38.5) | 9193 (32.9) | 1344 (36.1) |
| 5–9 | 18 080 (9.4) | 12 304 (10.8) | 3163 (6.7) | 2300 (8.2) | 313 (8.4) |
| 10–14 | 6468 (3.4) | 4059 (3.6) | 1344 (2.8) | 989 (3.5) | 76 (2.0) |
| 15–19 | 7319 (3.8) | 4126 (3.6) | 1755 (3.7) | 1327 (4.7) | 111 (3.0) |
| 20–29 | 28 242 (14.6) | 16 769 (14.7) | 6362 (13.4) | 4658 (16.7) | 453 (12.2) |
| 30–39 | 21 864 (11.3) | 12 398 (10.8) | 5305 (11.2) | 3742 (13.4) | 419 (11.3) |
| 40–49 | 12 041 (6.2) | 6203 (5.5) | 3359 (7.1) | 2189 (7.8) | 290 (7.8) |
| 50–59 | 5968 (3.1) | 2739 (2.4) | 1964 (4.1) | 1084 (3.9) | 181 (4.9) |
| 60–69 | 6398 (3.3) | 2466 (2.2) | 2581 (5.4) | 1123 (4.0) | 228 (6.1) |
| ≥70 | 7369 (3.8) | 2403 (2.1) | 3314 (7.0) | 1343 (4.8) | 309 (8.3) |
| Female sex | 105 035 (52.5) | 60 153 (51.1) | 27 184 (55.3) | 15 523 (53.3) | 2175 (56.1) |
| **Season of attendance** | | | | | |
| Rainy (Oct-Mar) | 103 798 (51.9) | 62 296 (52.9) | 25 008 (50.8) | 14 406 (49.4) | 2088 (53.8) |
| Cold and dry (Jun-Aug) | 47 189 (23.6) | 27 884 (23.7) | 11 385 (23.1) | 7129 (24.5) | 791 (20.4) |
| Transitional (Apr-May, Sep) | 48 893 (24.5) | 27 494 (23.4) | 12 791 (26.0) | 7606 (26.1) | 1002 (25.8) |
| Febrile* | 21 924 (17.5) | 16 149 (21.7) | 1736 (5.7) | 3543 (19.8) | 496 (19.2) |
| **Diagnostic** | | | | | |
| Gastroenteritis | 30 352 (15.2) | 21 008 (17.9) | 7816 (15.9) | 1256 (4.3) | 272 (7.0) |
| Sexually transmitted infection | 10 597 (5.3) | 9149 (7.8) | 545 (1.1) | 813 (2.8) | 90 (2.3) |
| Acute respiratory infection | 21 381 (10.9) | 16 290 (13.8) | 4119 (8.4) | 495 (1.7) | 477 (12.3) |
| Pneumonia | 20 889 (10.5) | 17 143 (14.6) | 914 (1.9) | 2673 (9.2) | 159 (4.1) |
| Tonsilitis | 13 940 (7.0) | 13 271 (11.3) | 210 (0.4) | 261 (0.9) | 198 (5.1) |
| Skin and soft-tissue infection | 7861 (3.9) | 4917 (4.2) | 922 (1.9) | 1918 (6.6) | 104 (2.7) |
| Upper respiratory tract infection | 7777 (3.9) | 6996 (5.9) | 594 (1.2) | 114 (0.4) | 73 (1.9) |
| Trauma | 11608 (5.8) | 2776 (2.4) | 3250 (6.6) | 5509 (18.9) | 73 (1.9) |
| Musculoskeletal pain | 5274 (2.7) | 527 (0.4) | 4055 (8.2) | 438 (1.5) | 254 (6.5) |
| Hypertension | 5423 (2.6) | 286 (0.2) | 3226 (6.6) | 1753 (6.0) | 158 (4.1) |
| Other diagnoses** | 65 818 (32.9) | 26 072 (22.1) | 23 690 (48.2) | 14 030 (48.1) | 2026 (52.2) |

*Missing age: 6982, missing sex: 3, *74464 without temperature recorded in the medical records; §age does not have a normal distribution in the population and is presented as median with IQR. **A listing of other diagnoses is presented in S1 Table B in S1 File.*

(39.4%; 65,825/166,858), ciprofloxacin (10.3%; 17,162/166,858), metronidazole (9.4%; 15,681/166,858), ceftriaxone (5.9%; 9,802/166,858), doxycycline (5.4%; 9,067/166,858), and erythromycin (5.1%; 8,479/166,858). Fig 2 shows antibiotics prescribed according to age.

Among the 29,141 patients referred to hospital, 3,299 (11.3%) were prescribed antibiotics prior to referral. Antibiotics commonly prescribed were penicillin (2,314/3,299, 70.1%) and gentamicin (1,612/3,299, 48.9%). These were given in combination to 1,551/3,299 (47.0%) patients.

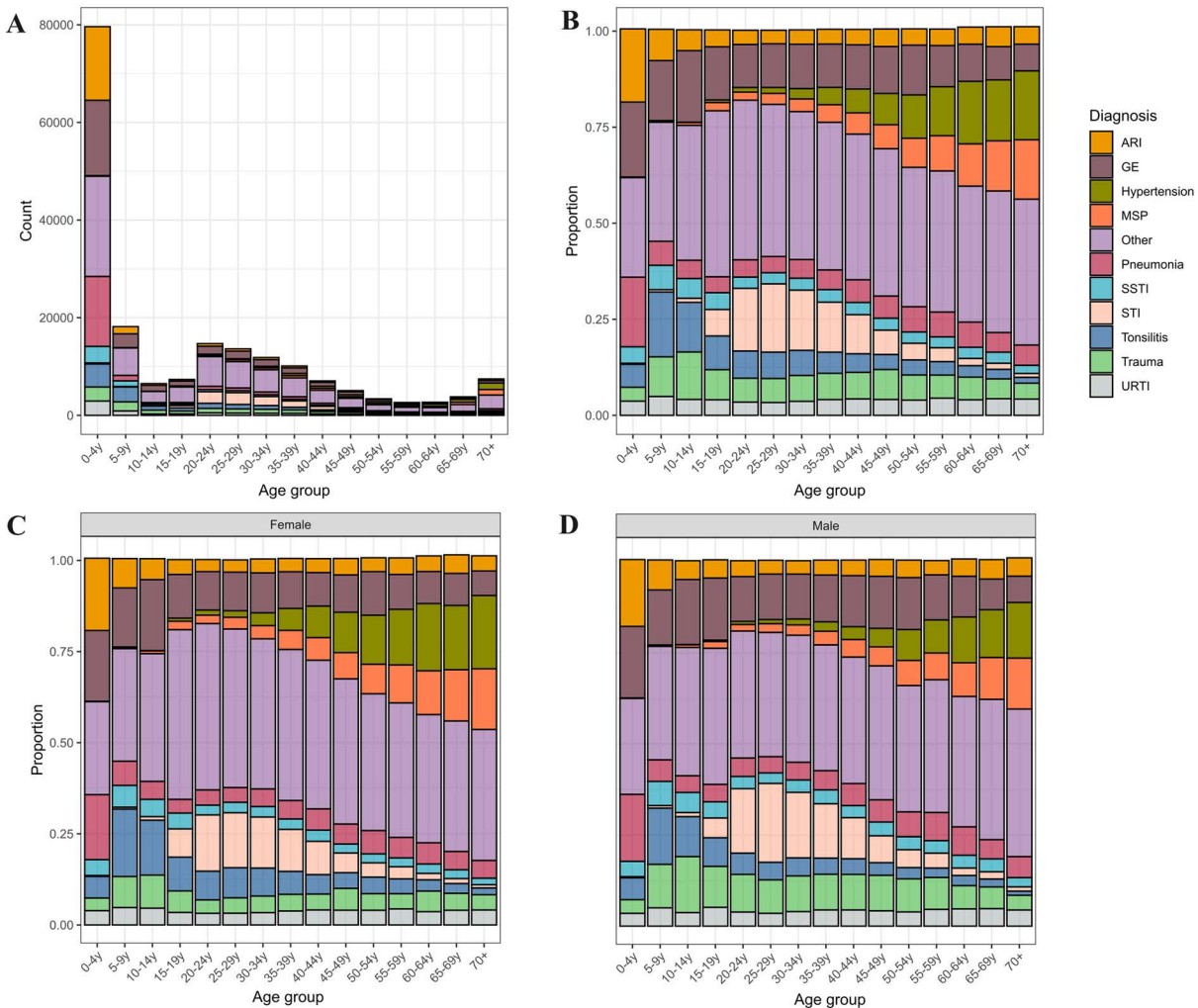

**Fig 1. Diagnoses according to age group: number with diagnosis (A); proportion with diagnosis (B) and according to age group and sex (C, D).** *ARI: acute respiratory infections; GE: gastrointestinal infections; MSP: musculoskeletal pain; SSTI: skin and soft-tissue infections; STI: sexually transmitted infections; URTI: upper respiratory tract infection. The other category encompasses: allergies, animal bites, COVID-19, diabetes mellitus, eye infectious, unclear fevers, malaria, mental health conditions, peptic ulcer disease, pregnancy-associated conditions, rash, tuberculosis, typhoid fever, urinary tract infections, etc. (specific diagnoses were usually recorded).*

## Antibiotic prescribing

Among those who were prescribed antibiotics and not referred, 70.6% (83,034/117,674) were prescribed 'Access' and 29.3% (34,472/117,674) 'Watch' group antibiotics. Ciprofloxacin was the main antibiotic prescribed from the 'Watch' group (49.8%; 17,162/34,472 of prescriptions). Of patients with 'Watch' antibiotic prescriptions, 34.5% (11,898/34,472), 24.2% (8,325/34,472) and 7.3% (2,513/34,472) were diagnosed with gastroenteritis, sexually transmitted infections, and skin and soft tissue infections, respectively. No patients were prescribed 'Reserve' antibiotics.

90.3% (55,746/61,706) of patients with respiratory tract infections were prescribed antibiotics of whom 89.5% (49,893/55,746) received amoxicillin. Among patients with pneumonia, 94.9% (17,144/18,057) were prescribed antibiotics; however, 92.2% (6,996/7,590) of those who

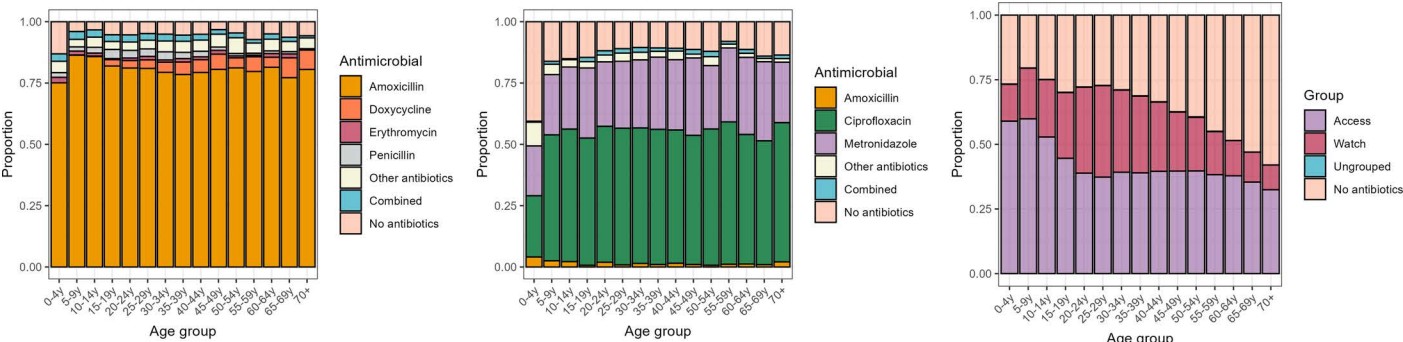

**Fig 2. Antibiotics prescribed: for respiratory tract infections\* (A); for gastroenteritis (B); and overall, according to the AWaRe classification (C).** *respiratory tract infections included the following diagnoses: upper respiratory tract infection (URTI), lower respiratory tract infection (LRTI), respiratory tract infection (RTI); acute respiratory infection (ARI), bronchitis, pneumonia, sinusitis, pharyngitis, tonsilitis, otitis, and COVID-19. Very few patients were prescribed antibiotics which did not belong to the AWaRe classification.*

were diagnosed with upper respiratory tract infections also received antibiotics. Patients with gastroenteritis received antibiotics in 72.9% (21,008/28,824) of consultations (Table 2, Fig 2).

## Patient factors associated with antibiotic prescriptions

Factors associated with antibiotic prescriptions were gender, age and recorded fever. Overall, females were 15% (95%CI 10–21) less likely to receive antibiotic prescriptions than men although there were no substantial differences for the most common diagnoses (S1 Fig F in S1 File). Antibiotic prescription rates decreased with increasing age (Fig 3 and S1 Fig G in S1 File). Patients with fever were 4.6 times (95%CI 4.3-4.8) more likely to receive antibiotics (Fig 3, S1 Table C in S1 File). There was some month-on-month variation in antibiotic prescriptions (S1 Table D in S1 File).

## Discussion

Limited data exist on the reasons for primary clinic visits and antimicrobial use in Zimbabwe. Our study provides valuable insights into outpatient presentations and antimicrobial prescriptions with the goal of informing policies and interventions for optimising antimicrobial use in this setting. This study collected routine data from eight primary care clinics serving a population of more than 700,000 over seven years, making these findings generalisable for this setting. Most outpatient presentations resulted in antibiotic prescriptions, of which 70% belonged to the 'Access' group which is lower than the suggested 90% 'Access' antibiotics for primary care [14]. Most 'Watch' group antibiotics were prescribed for diarrheal diseases and sexually transmitted infections. As expected, 'Reserve' antibiotics were not prescribed given that antibiotics belonging to this group are costly, often require intravenous administration, and are not usually available in the primary care setting.

High antibiotic prescription rates, ranging from 78% to 97% for respiratory infections have been reported from other low-resource settings [15,16]. This was also found in our study with antibiotics being commonly prescribed for respiratory infections in all age groups. Even for upper respiratory infections, antibiotics were overprescribed, with >90% of patients receiving a prescription. Gastroenteritis was an extremely common cause of presentation across all age groups and 70% of diarrheal episodes were treated with antibiotics, although antibiotics are not indicated for most diarrhoea cases. Because the medical records did not contain detailed clinical information on presentation and potential comorbidities, we were unable to ascertain

**Table 2. Antibiotic prescriptions among outpatients. Patients referred to hospital were excluded as they may have been prescribed antibiotics at the hospital. Patients with unknown referral status were also excluded.**

| Indicator | Denominator | Percent | n/denominator |
|---|---|---|---|
| **Overall antibiotic prescription** | | | |
| Antibiotic prescriptions | Total patients | 70.5 | 117 674/166 858 |
| Access antibiotics prescriptions | Patients with antibiotics | 70.6 | 83 034/117 674 |
| Watch antibiotics prescriptions | Patients with antibiotics | 29.3 | 34 472/117 674 |
| **Respiratory tract infections (RTI)** | | | |
| Presentations for any acute RTI | Total patients | 37.0 | 61 706/166 858 |
| RTI prescribed antibiotics | Total with RTI | 90.3 | 55 746/61 706 |
| RTI (any) prescribed amoxicillin | Total with RTI who were prescribed antibiotics | 89.5 | 49 893/55 746 |
| RTI prescribed Access antibiotics | Total with RTI who were prescribed antibiotics | 94.5 | 52 686/55 746 |
| RTI prescribed Watch antibiotics | Total with RTI who were prescribed antibiotics | 5.5 | 3058/55 746 |
| RTI prescribed antibiotics according to season | Total with RTI according to season | | |
| Rainy | | 90.3 | 28 654/31 722 |
| Dry | | 91.3 | 14 365/15 731 |
| Transitional | | 89.3 | 12 727/14 253 |
| Upper respiratory infections prescribed antibiotics | Patients with upper respiratory tract infections | 92.2 | 6996/7590 |
| Pneumonia prescribed antibiotics | Total with pneumonia | 94.9 | 17 143/18 057 |
| **Gastroenteritis** | | | |
| Presentations for acute gastroenteritis | Total patients | 17.3 | 28 824/166 858 |
| Gastroenteritis prescribed antibiotics | Total patients with gastroenteritis | 72.9 | 21 008/28 824 |
| Febrile§ gastroenteritis prescribed antibiotics | Total patients with febrile gastroenteritis | 85.3 | 2223/2606 |
| Non-febrile gastroenteritis prescribed antibiotics | Total patients with non-febrile gastroenteritis | 67.6 | 11 004/16 280 |
| Gastroenteritis prescribed Access antibiotics | Total patients with gastroenteritis who were prescribed antibiotics | 42.9 | 9002/21 008 |
| Gastroenteritis prescribed Watch antibiotics | Total patients with gastroenteritis who were prescribed antibiotics | 56.6 | 11 898/21 008 |
| Gastroenteritis prescribed antibiotics according to season | Total with gastroenteritis according to season | | |
| Rainy | | 75.0 | 12 019/16 017 |
| Dry | | 69.9 | 4380/6264 |
| Transitional | | 70.4 | 4609/6543 |
| **Skin and soft tissue infections (SSTI)** | | | |
| Presentations for SSTI | Total patients | 3.5 | 5839/166 858 |
| SSTI prescribed antibiotics | Total patients with an SSTI | 84.2 | 4917/5839 |
| SSTI prescribed Access antibiotics | Total patients with an SSTI receiving antibiotics | 48.9 | 2403/4917 |
| SSTI prescribed Watch antibiotics | Total patients with an SSTI receiving antibiotics | 51.1 | 2513/4917 |
| **Urinary tract infections (UTI)** | | | |
| Presentations for UTI | Total patients | 1.8 | 3072/166 858 |
| UTI prescribed antibiotics | Total patients with UTI | 97.3 | 2988/3072 |
| UTI prescribed Access antibiotics | Total patients with UTI who were prescribed antibiotics | 50.8 | 1519/2988 |
| UTI prescribed Watch antibiotics | Total patients with UTI who were prescribed antibiotics | 48.4 | 1445/2988 |

*Excludes referrals to hospital, upper respiratory tract infections as diagnosed by the health care workers; § documented fever*

whether antibiotics were indicated for individual patients. Excessive antibiotic prescriptions (50–77%) for diarrheal disease have been reported in other settings in Africa [17,18]. Yet it is estimated that around 80–85% of children presenting with simple gastroenteritis [19] do not require antibiotic treatment. Pathogen distribution and thus requirement of antibiotic treatment is dependent on local epidemiology and outbreaks, HIV and malnutrition burden, and immunization programmes. 'Watch' antibiotics, mainly ciprofloxacin, were frequently

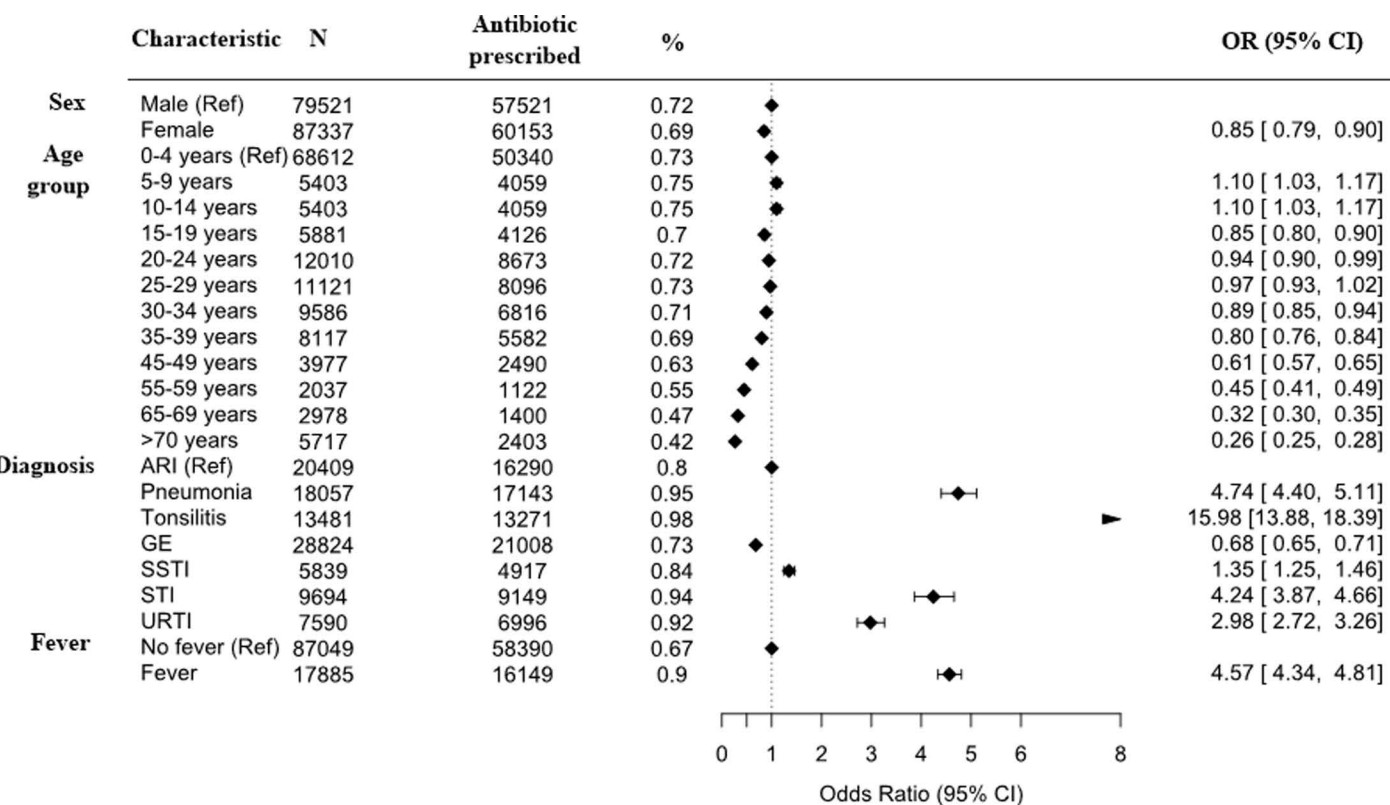

**Fig 3. Factors associated with prescribing of any antibiotic (unadjusted analysis).**

prescribed for gastroenteritis in our study. While the choice of antibiotic was often in accordance with the WHO AWaRe Antibiotic Book and the national EDLIZ (Essential Medicines List and Standard Treatment Guidelines for Zimbabwe) recommendations [13,20], antibiotics were likely unnecessary for the majority of episodes [19]. In addition, ciprofloxacin is the recommended treatment for typhoid fever and Harare was experiencing a typhoid outbreak during the time covered by the data collection period. The difficulties in typhoid fever diagnosis can lead to considerable overprescribing with between three and 25 unnecessary antibiotic prescriptions for each true typhoid case, depending on the setting [21]. For skin and soft-tissue infections, 'Watch' antibiotics were also frequently prescribed. This is due to the recommendation for prescribing erythromycin for skin and soft tissue infections in the EDLIZ guidelines [20], as well as amoxicillin-clavulanic acid and cefalexin being largely unavailable in Zimbabwe.

Sexually transmitted infections were particularly commonly diagnosed among younger adults where one out of six individuals received this diagnosis. Sexually transmitted infections are diagnosed and managed syndromically in Zimbabwe leading to both antibiotic overtreatment and undertreatment. A study conducted in Zimbabwe among individuals aged 16–24 years who underwent testing for *Chlamydia trachomatis* and *Neisseria gonorrhoeae* found that almost 20% of women and 10% of men had a positive test for sexually transmitted infections. However, only 40% of those reporting symptoms of sexually transmitted infections had a positive test, and most of those testing positive were asymptomatic [22]. These findings highlight the need for point-of-care testing for sexually transmitted infections in order to optimise antibiotic use and reduce transmission.

Severity of the conditions with which patients present may explain the high frequency of prescribed antibiotics and the high rates of referrals in this study. Patients with pneumonia were frequently referred likely due to the high proportion of young children among the presentations, the high burden of HIV in the population and the lack of access to diagnostics. Children under the age of five with pneumonia and diarrheal diseases were more commonly referred due to the increased risks of severe outcomes in this age group. In the absence of diagnostics to aid in decision-making and safety-netting should the patient deteriorate, health care workers have to strike a balance between ensuring access to antibiotics and rationalising antibiotic prescriptions [23]. While most respiratory and gastrointestinal infections are mild and do not require antibiotics, they also represent two of the common causes of death in young children [24]. Further, at the time of this study, several districts in Harare were experiencing a severe typhoid fever outbreak with almost 20% of blood cultures from febrile outpatients positive for *Salmonella* Typhi [25]. Given the high mortality and risk of severe complications associated with untreated typhoid fever, it is not surprising that antibiotic overtreatment is common [21] whenever this infection is suspected. Zimbabwe has been facing considerable socio-economic challenges and rapid inflation leading to out-migration of skilled staff including health care workers. This has led to understaffed clinics, high workloads and incomplete recording of consultations. Further, industrial action by health care workers, repeated outbreaks of cholera and typhoid, and more recently the COVID-19 pandemic have severely impacted health care provision further limiting access to health care.

The high rates of antibiotic prescribing are largely due to diagnostic uncertainties [26] and limited access to diagnostics resulting in scarce data on the local burden of infections and the prevalence of AMR. The lack of data leads to uncertain and inaccurate guidance about when and which antibiotics to use, to missed opportunities for informing public health interventions and to uncertainties in antibiotic prescribing on an individual level. While diagnostic uncertainty is common, only 19% of people living in LMIC have access to diagnostics at primary-care level (except tests for HIV and malaria for which there have been considerable investments facilitating the availability of rapid, reliable, and easy to use point-of-care tests) [27]. In Zimbabwe, access to microbiology services is also limited [28]. Studies from Asia and Africa have shown the potential of point-of-care diagnostics to guide prescribing and reduce antibiotic use. In Vietnam, C-reactive protein testing in ambulatory outpatients with respiratory illnesses reduced antibiotic prescriptions by a third [29]. Similarly, in a study conducted in West and Central Africa, the use of a diagnostic algorithm based on point-of-care testing for common infections reduced antibiotic prescriptions by 23% among children presenting with fever and respiratory symptoms [30]. These findings suggest that simple, inexpensive tests, ideally which can be done at the point-of-care, have the potential to reduce unnecessary prescriptions and optimise therapy in LMIC.

Further system-wide interventions such as immunizations and improving access to safe water and sanitation would also decrease the burden of infectious diseases. Vaccinations protecting against common respiratory infections (respiratory syncytial virus [RSV], influenza and pneumococcus) and gastrointestinal pathogens (such as rotavirus and *S.* Typhi) may also be effective in reducing the burden of infectious diseases and result in in reduced antibiotic prescriptions [31]. Despite the high access to safe water and sanitation in urban areas reported by the Zimbabwean census for 2022 [11], communities often experience severe water shortages resulting in repeated outbreaks of typhoid, cholera and diarrhoeal diseases [32,33]. Training initiatives on antibiotic prescribing may also play a role in improving antibiotic stewardship in Harare. In a study conducted in the same clinics from where the data were collected, health care workers reported that while they use the national guidelines to guide their practice, only a third had received training during the previous year on antibiotic prescribing [34].

The African continent is undergoing a rapid population growth and urbanisation with its population expected to triple by 2100 [35]. Rapid urbanisation might increase the numbers of people living in poor conditions, impact on their access to water and sanitation infrastructure and to preventative programmes thus increasing the risk of infections in crowded urban communities. Additionally, unprecedented declines in child mortality, increases in life expectancy, and lower fertility rates in Southern Africa have resulted in the most significant demographic transition in history. The population of older people in Southern, Western and Eastern Africa is projected to reach 163 million by 2050 [36]. Demographic shifts often go along with socio-economic changes resulting in changing patterns of morbidity from communicable to non-communicable diseases with substantial overlaps in these conditions. This pattern is starting to emerge in Zimbabwe as evidenced by the age-stratified disease presentations. Presentations of infectious diseases such as respiratory and gastrointestinal infections are more severe in people suffering from chronic diseases such as diabetes, cardiovascular diseases, chronic lung disease. While respiratory and gastrointestinal infections were the most common cause of presentation to primary care in children under the age of five, presentations by older age-groups for both infections and non-communicable conditions were frequent. Stewardship efforts might be challenged in these circumstances because of the presence of comorbidities influencing drug choice, polypharmacy, and the higher risk of infections and severe outcomes. Older children and adolescents presented to the clinics less frequently likely due to a lower burden of disease in the population and limitations in access because of having to pay clinic fees. Musculoskeletal pain was a common diagnosis particularly in older age groups. This comprises a larger group of conditions and represent a major cause of disability worldwide [37,38]. Of particular concern is that older people had relatively low hospital referral rates despite care being free for those over 65 years.

## Limitations

Our study is limited by its retrospective approach to data collection, the limited number of variables for which data were collected and incomplete recording often experienced in routine paper-based clinical registers. Diagnosis and management practices may have differed between health care workers and across clinics. Patients who accessed care in the private system and had antibiotics dispensed directly from pharmacies without a prescription were not captured. Although this practice was not common in Zimbabwe previously [39], it has likely increased due to economic constraints. Further, data on antibiotic dosages and duration of prescriptions and whether the prescription was actually collected by patients were not available, precluding the calculation of defined daily doses per population. This also prevents an assessment of whether doses and durations of antibiotic treatment were appropriate and in accordance with the guidelines. Detailed data on clinical presentation, co-morbidities and severity were not available to assess whether antibiotics were appropriate on a case-by-case basis. Due to insufficient information on clinical presentations, the retrospective design and large scale of the study, and recognizing the diagnostic difficulties health care workers face during consultations, we refrained from categorizing prescriptions on their appropriateness. Further, we could not determine recurrent attendances for the same individuals or conditions and whether patients followed referral recommendations for severe illnesses.

Excessive antibiotic prescribing accelerates the development and spread of AMR, rendering infections harder to treat. However, interventions that directly target prescribing may not be sufficient to reduce the AMR in LMIC and additional, wider system interventions such as improving water and sanitation infrastructure, better governance, and increasing healthcare expenditures may be more impactful [40,41]. These interventions are also key in reducing the burden of infections and as a result antibiotic use. AMR in sub-Saharan Africa is becoming

increasingly critical, especially as the region faces rapid population growth, urbanization, ageing, and a rise in chronic diseases. Despite these transitions, infectious diseases are anticipated to remain prevalent, leading to increased and often inappropriate antibiotic use. Without urgent action to improve access to diagnostics, preventative programmes and optimised prescribing, the region faces a crisis where AMR could severely undermine public health, threatening both lives and economic security.

## Conclusionss

This study, using a large dataset of routinely collected data from a low-resource settings, shows that infectious diseases remain a common reason for primary care presentation across all ages and, irrespective of presentation, almost two thirds of patients received antibiotic prescriptions. These data provide valuable insights into diagnoses and prescriptions across age groups which can be valuable for policy makers in this setting. Our findings highlight the need for diagnostics in primary care and antibiotic stewardship in this setting to optimise patient management, reduce unnecessary antibiotic prescriptions, and curb the increase in AMR.

## Supporting information

**S1 File. Table A** Number of presentations according to year and clinic.**Table B** Other diagnoses (includes all patients irrespective of hospital referral who did not receive any other diagnosis). **Table C** Characteristics of patients according to recorded temperature. **Table D** Antibiotic prescribing according to month of presentation. **Fig A** Map of Harare representing the clinics included in the study. **Fig B** Comparison of population age structure according to census data and clinic presentations. **Fig C** Number of outpatient clinic visits for the main diagnoses according to age group and sex. **Fig D** Proportions referred to hospital among outpatients with specific diagnoses. **Fig E** Referrals to hospital according to age group. **Fig F** Proportions prescribed antibiotics for the main diagnoses by sex. **Fig G** Factors associated with Watch antibiotic prescribing.
(DOCX)

## Acknowledgments

The authors would like to acknowledge the research staff who collected the data and the nurses from the polyclinics of Harare.

## Author contributions

**Conceptualization:** Ioana D. Olaru, Christian Bottomley, Justin Dixon, Thomas C Darton, Katharina Kranzer.

**Data curation:** Ioana D. Olaru, Fadzaishe Mhino, Tsitsi Bandason, Chipo E. Mpandaguta.

**Formal analysis:** Ioana D. Olaru, Christian Bottomley, Katharina Kranzer.

**Funding acquisition:** Ioana D. Olaru, Justin Dixon, Katharina Kranzer.

**Investigation:** Ioana D. Olaru, Rudo MS Chingono, Chipo E. Mpandaguta, Karlos Madziva, Rashida A. Ferrand, Justin Dixon, Thomas C Darton, Katharina Kranzer.

**Methodology:** Ioana D. Olaru, Christian Bottomley, Justin Dixon, Thomas C Darton, Katharina Kranzer.

**Project administration:** Ioana D. Olaru, Rudo MS Chingono, Thomas C Darton, Katharina Kranzer.

**Resources:** Ioana D. Olaru, Fadzaishe Mhino, Tsitsi Bandason, Chipo E. Mpandaguta, Rashida A. Ferrand, Michael Vere, Prosper Chonzi, Shungu Munyati, Thomas C Darton, Katharina Kranzer.

**Supervision:** Ioana D. Olaru, Rudo MS Chingono, Fadzaishe Mhino, Celia Gregson, Christian Bottomley, Chipo E. Mpandaguta, Rashida A. Ferrand, Michael Vere, Prosper Chonzi, Shungu Munyati, Justin Dixon, Thomas C Darton, Katharina Kranzer.

**Validation:** Ioana D. Olaru.

**Visualization:** Ioana D. Olaru, Katharina Kranzer.

**Writing – original draft:** Ioana D. Olaru, Thomas C Darton, Katharina Kranzer.

**Writing – review & editing:** Ioana D. Olaru, Rudo MS Chingono, Fadzaishe Mhino, Celia Gregson, Christian Bottomley, Tsitsi Bandason, Chipo E. Mpandaguta, Karlos Madziva, Rashida A. Ferrand, Michael Vere, Prosper Chonzi, Shungu Munyati, Justin Dixon, Thomas C Darton, Katharina Kranzer.

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
