## [Decision Letter · Decision Letter 0]

14 Jan 2025

PGPH-D-24-02556

High burden of infectious diseases and antibiotic prescribing among patients presenting to primary care in Harare, Zimbabwe

Dear Dr. Olaru,

Thank you for submitting your manuscript to PLOS Global Public Health. After careful consideration, we feel that it has merit but does not fully meet PLOS Global Public Health’s publication criteria as it currently stands. Therefore, we invite you to submit a revised version of the manuscript that addresses the points raised during the review process.

***Please note that reviewer 1 comments are contained in an attachment, reviewer 2 comments in text below.**

We look forward to receiving your revised manuscript.

Kind regards,

Gemma Lea Saravanos

Academic Editor

Journal Requirements:

2. We noticed that you used "unpublished data" in the manuscript. We do not allow these references, as the PLOS data access policy requires that all data be either published with the manuscript or made available in a publicly accessible database. Please amend the supplementary material to include the referenced data or remove the references.

Additional Editor Comments (if provided):

Reviewers' comments:

Reviewer's Responses to Questions

**Comments to the Author**

1. Does this manuscript meet PLOS Global Public Health’s publication criteria ? Is the manuscript technically sound, and do the data support the conclusions? The manuscript must describe methodologically and ethically rigorous research with conclusions that are appropriately drawn based on the data presented.

Reviewer #1: Partly

Reviewer #2: Yes

2. Has the statistical analysis been performed appropriately and rigorously?

Reviewer #1: No

Reviewer #2: Yes

3. Have the authors made all data underlying the findings in their manuscript fully available (please refer to the Data Availability Statement at the start of the manuscript PDF file)?

Reviewer #1: Yes

Reviewer #2: Yes

4. Is the manuscript presented in an intelligible fashion and written in standard English?

Reviewer #1: Yes

Reviewer #2: Yes

5. Review Comments to the Author

Reviewer #1: The manuscript addresses a serious public health concern and an provide enough information to the domain. But it needs editing for clarity, and rigor in describing the methodology results and Discussion before it is ready for publication.

Reviewer #2: This is a well written interesting paper, presenting an overview of antibiotic use in selected Harare health facilities, using routine clinical data.

Such data are relatively rare to access in most of sSA, due to a lack of standardized digitally available surveillance data, but in view of increasing AMR and ABU (including Watch) highly relevant as a basis for public health policies and interventions to protect antibiotics and public health.

I still have a few questions and comments:

Background:

-can data on prevalence of HIV and malnutrition be added

Methods:

-please clarify how the clinics were selected for inclusion, and to what extent this can have biased results? Eg access to care, socio-economic background, staffing, etc may vary by facility?

-can the authors clarify inclusion criteria? It is mentioned ‘acute illness’, but ‘acute’ is not defined. Eg it is a bit surprising this would include hypertension, while on the other hand it is likely to miss out on many other NCDs, impacting on the observations on the double burden of disease. Is there information on the total number of all consultations (acute and not acute)?

-please clarify the referral process: it is mentioned that patients were ‘recommended to refer to hospital’: does this mean some may have gone, and others not; depending upon self-referral? Also,? it is mentioned that direct referrals were excluded from the analysis of antibiotics, but does this also relate to direct referrals who already received antibiotics?

-to what extent does the more generic category acute respiratory infection overlap with more specific categories for pneumonia and upper respiratory tract infections? Are these 3 categories used consistently over time, place and patient group?

-the diagnostic group ‘others’ is very large (a third to a quarter of all diagnosis), more details should be provided (the minimum would be a distinction into infectious/non-infectious). It is stated that it consists of diagnosis mentioned for less than 2%, but it might be possible to group some into larger categories (similar as presumably done for most of the specified categories, such as GE, STI, SSTI, trauma, ). Eg malaria and unclear fevers are rather common presentations in many sSA settings, which can trigger antibiotic use, and it would be relevant to show their contribution, and clarify to what extent malaria diagnostics were available/used. Also, this is not consistent throughout: in table 1 ‘rash’ is included as a separate category, in table 2 ‘urinary tract infections’; while in both tables ‘others’ is missing

-data were captured from retrospective paper registers over a long period of time. Please provide some more information to what extent quality and completeness may have varied over time, and between centres

-please clarify the study period. It is stated data were collected between Jan 2020 and April 2022, capturing the period from Jan 2016 till December 2022; this does not seem possible

Results:

-fever was not recorded for over a third of patients, but this may not have been random? Eg for patients presenting with hypertension or musculoskeletal pains, this may be less indicated; for presumed infectious diseases, this may be one of the indicators for antibiotic use. Can data be presented on antibiotic use of the main categories with and without fever? (in table 2 includes this for GE only)

-to explore potential differences between men and women in antibiotic use, it would be useful to provide a sensitivity analysis which excludes children under the age of 10, as they represent about half of all patients, and could mask gender differences among adults

-table 1: please include all diagnostic categories (figure 1 provides information on 11, not 8 categories); and clarify the category ‘rash’ as before it was stated this was part of ‘others’, in line with it comprising 1.8% of all diagnoses (‘others’ is one of the categories still to be included in the table, preferably with some more subdivision; see above). The text states that acute pharyngitis was the fourth most common diagnosis, which is captured as tonsillitis in table and figure, please be consistent. -check for consistency of the provided numbers. Eg it is stated several times that 199,980 consultations were recorded over the study period, but on page 9, 195,999 is given as the total (of which 29141 were referred (or recommended to refer?))

Discussion:

-it is remarkable that hardly any seasonality was observed; can the authors comment?

Minor:

-check spelling (eg in abstract: findings: .. patients not refereed to hospital ..

6. PLOS authors have the option to publish the peer review history of their article (what does this mean? ). If published, this will include your full peer review and any attached files.

**Do you want your identity to be public for this peer review?** For information about this choice, including consent withdrawal, please see our Privacy Policy .

Reviewer #1: No

Reviewer #2: No

---

## [Editor Report · Decision Letter 1]

10 Mar 2025

Infectious diseases burden and antibiotic prescribing patterns among primary care patients in Harare, Zimbabwe – a cross-sectional analysis

PGPH-D-24-02556R1

Dear Dr Olaru,

We are pleased to inform you that your manuscript 'Infectious diseases burden and antibiotic prescribing patterns among primary care patients in Harare, Zimbabwe – a cross-sectional analysis' has been provisionally accepted for publication in PLOS Global Public Health.

Best regards,

Gemma Lea Saravanos

Academic Editor